

# Contrasting early- and late-Holocene vegetation and wildfire regimes in a high-value drinking water supply area, Canada

Daniel R. Horrelt[1,2], Kendrick J. Brown[1,2,3], Nicholas Conder[4], John A. Trofymow[1,4], Christopher Bone[1]

[1] Natural Resources Canada, Northern Forestry Centre, Edmonton, AB, T6H 3S5, Canada
5  [2] Department of Geography, University of Victoria, BC, V8P 5C2, Canada
[3] Department of Earth, Environmental, and Geographic Sciences, University of British Columbia Okanagan, Kelowna, BC V1V 1V7, Canada
[4] Natural Resources Canada, Pacific Forestry Centre, Victoria, BC, V8Z 1M5

*Correspondence to*: Daniel R Horrelt (rosshorrelt@uvic.ca)

10  **Abstract.** Paleoenvironmental reconstructions of past ecosystems and fire regimes can strengthen interpretations of modelled future fire environments. In this study, sediment cores from four lakes in a high-value water supply area on southern Vancouver Island, Canada, are used to compare climate, vegetation, and fire along a regional east-west precipitation gradient between warm-dry early- and cool-moist late-Holocene intervals. Results indicate that inferred precipitation was lower in the past, with more open-canopy xeric Douglas-fir forests compared to present-day closed-canopy mesic western hemlock and cedar forests. 15  Overall, the wettest and western-most site experienced the greatest change, with more frequent early-Holocene fires yielding to longer fire return intervals in the late-Holocene. This implies that northern coastal temperate rain shadow forests, currently experiencing little fire, may become more vulnerable in the future. It also highlights susceptibility to fire regime shifts consistent with regional observations and models suggesting current and future increases in extreme fire disturbance.

## 1 Introduction

20  Climate change is driving an increase in global wildfire activity with significant economic, social, and environmental consequences (Liu *et al*., 2010; Wang *et al*., 2023; MacCarthy *et al*., 2024; Bowman *et al.,* 2017; Mattioli et al., 2022; Clarke *et al*., 2023; Chen *et al*., 2024, Burton *et al*., 2024). Temperate coniferous forests, typically located in coastal and mountainous regions and characterized by cool-wet winters and warm dry summers, are not exempt. Over the last two decades, these forests experienced a disproportionate abundance of extreme fire events and are forecast to see a further 25% increase in conditions 25  conducive to fire by the end of the century (Senande-Rivera *et al*., 2022; Cunningham *et al*., 2024).

In western North America, area burned and frequency of large fires across a range of forest types are increasing, with models forecasting more extreme fire weather over the coming decades (Flannigan *et al*., 2005; Westerling *et al.,* 2006; Wang *et al*., 2015, 2017; Hanes *et al*., 2019). For example, the 2023 wildfire season in the western Canadian province of British Columbia (BC), was the worst on record. Within BC, an estimated 2.84 million hectares (ha) burned, multiple lives were lost, and 30  approximately CAN$817 million was spent on wildfire suppression (BC Wildfire Service, 2024). While fire activity in coastal BC was historically limited by climate, ignition potential, and vegetation structure, recent notable wildfire seasons (i.e. 2017,



2018, 2021 and 2023) reflect a fire regime in transition (Parisien *et al*., 2023). In particular, the Coastal Fire Centre recorded 291 fires combusting 175,008 ha of forested land in 2018, and 365 fires burning 89,750 ha during the 2023 season (BC Wildfire Service, 2024). These trends, along with climate envelope modelling and fire simulations, are challenging assumptions that the moist forests in the Pacific region of North America are less susceptible to fire disturbance (Wang *et al*., 2017; Dye *et al*., 2023) and raises new questions about the significance of shifting fire regimes in areas previously overlooked.

The linkages between climate and fire systems are well established (Overpeck, 1990; Agee, 1993; Whitlock *et al*., 2003; Power *et al*., 2008; Marlon *et al*., 2009; Fishcer *et al*., 2015), but the uncertainty produced by global climate models (GCM) and climate-fire non-stationarity complicates analyses of forest response to future projected warming (Littell *et al*., 2018; Dye *et al*., 2023). While predictive models offer key insights into wildfire risk and climate adaptation, they are limited by incomplete historical fire data, lack of fire records, and human land management practices such as fire exclusion (Westerling *et al*., 2006; Moritz *et al*., 2014; Hessburg *et al*., 2019). Furthermore, northern temperate forests often exhibit mixed-severity fire regimes and have highly variable fire return intervals with multiple extrinsic and intrinsic drivers of disturbance (Agee, 2005; Halofsky *et al*., 2018). Paleoecological investigations can help elucidate some of these issues by improving understanding of long-term shifts in climate and fire activity, contributing to the strength of interpretation and validation of models (Marlon, *et al*., 2009, 2020; Brown *et al*., 2019; 2022; 2023). Specifically, pollen sequences offer insights into vegetation and climate, while charcoal records can be used to reconstruct past fire regimes, extending insights beyond the comparatively short period of observed and satellite records (Brown and Power, 2013; Chevalier *et al*., 2020). None the less, the past does not provide a direct analogue for the future. Instead, it reveals how ecological systems and processes change in response to various drivers.

The early-Holocene (ca. 7000 – 11,700 cal. yrs BP) interval in coastal BC was warmer and drier compared to present (Mathewes and Heusser, 1981; COHMAP, 1988; Hebda, 1995; Rosenberg *et al.,* 2004; Brown *et al*., 2006). The expansion of shade intolerant Douglas-fir (*Pseudotsuga menziesii* (Mirb.) Franco) after 11,700 cal yrs BP, together with grasses (*Poaceae*), roses (*Rosaceae),* and bracken ferns (*Pteridium*)*,* imply that open canopy forest prevailed, likely in response to both climate and frequent fire disturbance (Cwynar, 1987; Brown *et al*., 2008; Lucas and Lacourse, 2013). Indeed, prevailing low streamflow, as a proxy for fire season length, suggests that fire seasons were likely of longer duration (Brown *et al*., 2008, 2019; 2023). In contrast, over the last 4000 years, comparatively cooler and moister climatic conditions prevailed, facilitating establishment of modern temperate forests, with changing fire regime. On Vancouver Island, fire return intervals (FRI) convey high spatial variability in the region, ranging from <100 years on drier eastern sites (Pellatt *et al*., 2015) to longer intervals (>1000 years) reported at sites in wetter settings (Brown and Hebda, 2002a; Gavin *et al.,* 2003b). With ongoing greenhouse gas emissions and climate warming, it is possible that some characteristics of the early-Holocene fire regime may soon remerge. Thus, the objectives of this study are to (1) examine changes in vegetation and fire disturbance across a regional climatic gradient in a northern coastal temperate rains shadow forest and (2) assess how fire regimes changed through time to determine if past conditions could reflect evolving trends. This study compares two distinct climate intervals using lake



sediment cores extracted from four sites located in a high value municipal drinking water supply area on Vancouver Island,
BC, Canada. Given that wildfire threatens the provision of drinking water for a substantial population in the area (ca. 430,000
people), the persistent-long term effects of forest and fire regime change is of interest to regional water purveyors. The general
applicability of this research within similar contexts will have universal appeal for practitioners developing wildfire risk
reduction and adaptation strategies.

## 1.1  Study setting

The Greater Victoria Water Supply Area (GVWSA) occupies 20,549 hectares (ha) of forested land (48.4°N to 48.6° N; -
123.5°W to -123.9°W; Fig. 1). It is composed of three watersheds (Goldstream, Sooke, and Leech) that are managed by the
Capital Regional District (CRD), a regional government for 13 municipalities, and protected by controlled access. Drinking
water is largely sourced from the Sooke Lake Reservoir, with minimal treatment (CRD, 2017). Local climate is strongly
influenced by proximity to the ocean and mountainous topography, where regular wet frontal low pressure systems in winter
are punctuated by persistent warm-dry conditions in the summer from blocking highs (Dermachi, 2011). Regarding vegetation,
BC uses a nested zonal system of biogeoclimatic ecological classification (BEC; Meidinger and Pojar, 1991), in which the
forests within the watersheds are classified as Coastal Western Hemlock (CWH). Two main BEC subzones are delineated by
an east-west precipitation gradient, with CWHxm (xeric maritime) in the drier east and CWHmm (moist maritime) in the
moister west (Table 1; Fig.1). Further east of the water supply, a third BEC subzone, Coastal Douglas-fir (CDF) occupies the
driest climate niche on Vancouver Island.

Four sites were selected for study based on their contrasting biogeoclimatic conditions (Table 1). Frog Lake is the southeastern
most study site and represents the driest and warmest climate, with the surrounding topography distinguished by south-
southwestern aspects, steep rocky bluffs below the site, and hummocky terrain above. Surrounding vegetation consists of
Douglas-fir (*P. menziesii*) and western hemlock (*Tsuga heterophylla* (Raf.) Sarg.), with western red cedar (*Thuja plicata* Donn
ex. D. Don). Salal (*Gaultheria shallon* Pursh*)*, Oregon grape (*Mahonia nervosa* (Pursh) Nutt.*)*, and red huckleberry (*Vaccinium
parvifolium* Sm. In Rees) dominate the shrub layer together with small amounts of oceanspray (*Holodiscus discolor* (Pursh)
Maxim.) and baldhip rose (*Rosa gymnocarpa* Nutt.). Bigleaf maple (*Acer macrophyllum* Pursh) and red alder (*Alnus rubra*
Bong.) are common in the area. The herb layer consists mainly of twinflower (*Linnaea borealis* ssp*. Longiflora* (Torr.) Hulten),
Vanilla leaf (*Achlys triphylla* (Sm.) DC.), sword fern (*Polystichum munitum* (Kaulf.)  Presl), and bracken fern (*Pteridium
aquilinum* (L.) Kuhn). Begbie Lake is the lowest elevation site, with the basin having the smallest surface area of all sites
investigated. It is located just north of the main Sooke Lake Reservoir. At this site, the topography is generally flat with a
marginal wetland around the lake giving rise to a hill with dominant southeastern aspects above the site. Vegetation is similar
to Frog Lake, though with Sweet gale (*Myrica gale* L.) present along the lake margin.






**Figure 1: Study area map of the GVWSA. (a) Regional overview and study area outline (red box), world hill shade (ArcGIS Map Service, 2024: Esri, Maxar, Airbus DS, USGS, NGA, NASA, CGIAR, N Robinson, NCEAS, NLS, OS, NMA, Geodatastyrelsen, Rijkswaterstaat, GSA, Geoland, FEMA, Intermap, and the GIS user community). (b) The GVWSA with site locations and BEC subzones (Government of BC, 2021). (c) Average total annual precipitation, 1991 – 2020 (Wang *et al.*, 2016). (d – g). Worley Lake,**
**Begbie Lake, Swanson Lake, and Frog Lake respectively, showing coring locations, surrounding topography (20 m contours, white lines) and scale (0 – 300 m), world imagery (ArcGIS Map Service, 2024: Esri, Maxar, Earthstar Geographics, and the GIS User Community).**



Swanson and Worley lakes lie within the moist CWHmm subzone. Swanson Lake is the shallowest lake studied and sits just
inside the Leech watershed near the western boundary of the Sooke where it represents the most eastern extent of CWHmm.
The site occurs on top of a prominent undulating ridge. A few small craggy features protrude above the site providing both
north and southwest facing local aspects. The ridge slopes steeply down from the site to the north and east. Primary tree species
include western hemlock, Douglas-fir, amabilis fir *(Abies amabilis* (Douglas ex Loudon) Douglas ex Forbes), and yellow cedar
(*Chamaecyparis nootkatensis* (D. Don) Oerst. ex D.P. Little). Lodgepole pine (*Pinus contorta* Douglas ex Loudon
var. *contorta)* is also notable at the site. The shrub layer is mainly Alaskan blueberry (*Vaccinium alaskaense* Howell*)* and salal
with five-leaved bramble (*Rubus pedatus* Sm.). Worley Lake is the western most site, as well as the largest and deepest lake.
The site sits on a bench surrounded by gentle, hummocky terrain and local topography that reflects both north and south
aspects. Vegetation is represented by western hemlock, Douglas-fir, amabilis fir, and yellow cedar and a shrub layer that
consists mainly of Alaskan blueberry, salal and other *Vaccinium* species. The herb layer is less well developed compared to
eastern sites, consisting mainly of vanilla leaf amongst dominant moss cover.

**Table 1: Characteristics of study lakes within the GVWSA. Locations, BEC subzones (BC Forest Analysis and Inventory Branch, 2024), elevations in metres above sea level (asl.), and annual precipitation (ppt) in millimetres (mm) from Climate BC (Wang *et al*., 2016).**

| Site | Coordinates (° N, ° W) | Watershed | BEC Zone/ Subzone | Elevation (m asl) | Annual ppt (mm) | Lake size (ha) | Depth (m) |
|---|---|---|---|---|---|---|---|
| Frog Lake | 48.48, 123.59 | Goldstream | CWHxm | 460 | 1,587 | 2.31 | 7.9 |
| Begbie Lake | 48.58, 123.68 | Sooke | CWHxm | 189 | 1,511 | 0.77 | 8.3 |
| Swanson Lake | 48.57, 123.74 | Leech | CWHmm | 774 | 2,396 | 1.65 | 5.0 |
| Worley Lake | 48.59, 123.81 | Leech | CWHmm | 809 | 2,884 | 3.56 | 10.6 |

## 2 Material and Methods

Sediment cores were recovered from the deepest part of each lake basin. Upper unconsolidated sediment was recovered using
a 50-cm long Kajak corer (Kajak *et al*., 1965), whereas a Livingstone piston corer (Wright, 1967; Livingstone, 1955) was used
to retrieve deeper sediment in 1 m increments. To establish chronologies, plant macrofossils were sent to the University of
Ottawa A.E. Lalonde AMS Laboratory for radiometric ($^{14}$C) age determination. Radiocarbon ages were calibrated to calendar
years using OxCal v4.4 (Ramsey, 2009) and the IntCal20 calibration curve (Reimer *et al*, 2020). A volcanic ash layer,
consisting of glass shards when viewed under a compound microscope, was observed in all cores. Given its stratigraphic
location relative to other radiocarbon dates coupled with the known distributions of regional ash plumes, it is likely to be



Mazama ash (Egan *et al.*, 2015). The age-depth model was established using the *Bacon* package for *R* (V 4.1.1), which uses a Bayesian approach to estimate accumulation rates from radiocarbon dates through Markov Chain Monte Carlo (MCMC) iterations (Blaauw and Christen, 2011). Given that Mazama ash is an instantaneous event, its thickness was omitted from

modelling. Priors for the accumulation rate are modelled using a gamma distribution and memory (autocorrelation) is defined using a beta distribution (Fig. 2). Default parameters (Goring *et al.*, 2012) were applied to all models except Worley Lake where the acc.shape was decreased to reduce the peakedness of the prior distribution and produce a better fit.

Magnetic susceptibility profiles were developed for each core using a Bartington MS3 metre and MS2E core logging sensor (area of response 3.8 × 10.5 mm, depth of response 50% at 1 mm and 10% at 3.5 mm) recorded through Bartsoft PCv4.0

(Bartington Instruments, 2013).  Measurements were taken at 0.5 cm intervals for Kayak samples and at 1 cm intervals for the Livingston cores, applying regular temperature drift corrections at every fifth reading. Each section was repeat measured and plotted to validate the profile with offsets applied as necessary.

Pollen and spores (henceforth pollen) were analyzed using traditional methods (Moore *et al.*, 1991). For each targeted section of core, a 1cm$^3$ sediment sample was collected every 5 cm and sieved at 150 μm, retaining the finer component. An exotic

*Lycopodium* tablet was added as a spike to each sample (20,848 ± 1,546 spores tablet$^{-1}$; Department of Quaternary Geology, University of Lund, batch no. 1031). Pollen were extracted through chemical digestion involving HCl and acetolysis (Moore *et al.*, 1991), with residues mounted in glycerine in the resultant slides. A minimum of 300 grains per slide were tallied using a Fisher Scientific Micromaster compound microscope at 400-1000× magnification, with identification aided by published keys (Moore *et al.,* 1991; Kapp, 1969; and McAndrews *et al.*, 1973) and reference slides. Taxa that were not definitively

identified to genus or species were harmonized by family. Pollen counts were standardized using the *decostand* function in the *vegan* package for *R* 4.1.1 (Oksanen *et al.*, 2022) and expressed as percentage of pollen per sample. Pollen diagrams were plotted using the *Rioja Plot* V0.1-20 package (Juggins, 2023) with a subset (relative abundance <5%) of taxa removed. Using the complete pollen dataset, but excluding aquatics, taxa were classified as arboreal (AP) and non-arboreal (NAP) and grouped into fire-related functional categories (invaders, avoiders, evaders, endurers, resisters, and unknowns), which are indicative of

fire adaptations (Rowe, 1983; Agee, 1993; Wirth, 2005; Giuliano and Lacourse, 2023). Cumulative graphs of functional groups were also plotted using *Rioja Plot.*

Macrocharcoal (>150 μm) was quantified from sieved subsamples (3 cm$^3$) collected contiguously in 1 cm increments. Charcoal was defined as vegetative material altered by fire, resulting in black, opaque particles with vestigial cellular structure and submetallic lustre that was brittle upon manipulation (Brown and Power, 2013). The total number of charcoal fragments per

sample (fragments cm$^{-3}$) were tallied using a Leica stereomicroscope at 30 – 45× magnification. To infer the origin of burnt materials, charcoal fragments were classified into four functional categories (woody materials, graminoids, deciduous/herbaceous materials, and unclassified) based on morphology (Enache and Cumming, 2006; Mustaphi and Pisaric, 2014; Feurdean, 2021). Key distinguishing features included presence of tracheids and bordered pits in wood charcoal



(principally from conifers in the study area), rectangular cells and oval stomata in graminoid types, and diverging branches
and netted venation in deciduous/herbaceous types (Jensen *et al*., 2007).

Charcoal peak detection was conducted using *tapas* (Finsinger and Bonnici, 2022) representing an R script version of trend
and peak CharAnalysis (Higuera *et al*., 2009; 2011). CHAR (fragments cm$^{-2}$ yr$^{-1}$) records were resampled and binned to equally
spaced time intervals using median sample resolution, and subsequently decomposed using robust locally weighted scatterplot
smoothing to separate background from peak components. Smoothing window widths were defined for each record using
sensitivity screening. The peak component was identified by a local Gaussian mixture model and minimum-count test at a 0.95
threshold, with suitability evaluated using a signal-to-noise index (SNI; Kelly *et al*., 2011). An index of fuels combusted was
developed based on the ratio of graminoid to woody charcoal influx (GWI; Eq. (1)) where $CHAR_G$ and $CHAR_W$ represent the
interpolated charcoal accumulation rates for graminoid and woody materials, respectively. Values closer to one indicate higher
rates of wood burning. An overall fuel morphotype index (FMI) was then calculated for each site interval from the average
GWI (Eq. (2)).

**Eq. (1)**

$$GWI = 1 - \left( \frac{CHAR_G}{(CHAR_G + CHAR_W)} \right)$$

**Eq. (2)**

$$FMI = \frac{\sum_{i=1}^{n} GWI_i}{n}$$

*Tapas*-derived temporal distribution of peaks yielded estimates of fire frequency and FRIs, with the latter group-factored by
site and interval (early- vs. late-Holocene). The distributions of FRIs were tested and failed assumption for normality using a
Shapiro-wilk's test on residuals of a linear model. To test if FRIs differed between the early- and late-Holocene at each site
(dependent groupings), a non-parametric Wilcoxon test was performed using *rstatix* 0.7.2 (Kassambra, 2022).  A Kruskal-
Wallis test was performed across all independent groupings (comparing FRIs between sites) followed by Dunn's pairwise
comparisons.

Standardized and transformed charcoal influx was used to reduce site effects and variability between records and compare
biomass burning in the GVWSA (Marlon *et al*., 2013; Conedera *et al*., 2009; Marlon *et al*., 2016). Charcoal concentrations
were transformed into influx values, variance homogenized using a Box-Cox transformation, rescaled by minmax, and passed
to *Z* scores. The transformed data were presampled and binned in 20-year steps for base periods representing the early (11,000
– 8,000 cal yr BP) and late-Holocene (3000 – 0 cal yr BP). A composite curve was fit by bootstrap of site confidence intervals



using a local regression function with fixed half window width (smoothing parameter = 500). Individual site curves were generated from transformed, prebinned individual site data and fit using the same regression parameters.

Mean annual precipitation (MAP) was reconstructed using an index of xeric-adapted *P. menziesii* to hydric-affiliated *T. heterophylla (DWHI;* Brown *et al*., 2006*)*. Downcore estimates of MAP were subsequently generated using a regression model based on surface pollen spectra and PRISM gridded precipitation (Brown *et al*., 2006; Brown and Schoups, 2015; Daly *et al*., 1994; Hamann and Wang, 2005). Furthermore, mean annual precipitation (MAP, BIO12) and mean maximum monthly temperature (Tmax., BIO05) were extracted using *pastclim* 1.2 (Leonardi *et al.,* 2023) to plot the PALEO-PGEM (Barreto *et al*., 2023) and HadCM3 series' (Beyer *et al.,* 2020).These are presented alongside two chironomid-based estimates of mean July air temperature (MJAT) from lakes located 30 and 240 km from the GVWSA (Pellatt *et al*., 2000; Lemmen and Lacourse, 2018).

## 3 Results

### 3.1 Chronologies

Sediment cores from all lakes were principally comprised of brown gyttja with basal grey clay (total lengths [cm], Frog = 562, Begbie = 633, Swanson = 654, Worley = 474; Fig. 2). Age-depth models returned mean 95% confidence ranges of 492, 543, 501, and 585 years for Frog, Begbie, Swanson, and Worley lakes, respectively.





**Figure 2: Age-depth models for study sites in the GVWSA. (a) Worley, (b) Begbie, (c) Swanson, and (d) Frog lakes. Each panel shows**
**the calibrated ¹⁴C dates (blue), modelled mean age (red dash line), 95% confidence intervals (grey stippled area) and modelling**
**parameters. The horizontal grey line in each panel represents Mazama ash. Dashed brackets indicate targeted sampling intervals**
**for the early- and late-Holocene. Parameter settings are also presented in light grey text; acc.shape, is the prior used to model**
**accumulation rate, acc.mean is the prior for the mean in years/cm, mem.strength is the prior used to model memory, and mem.mean**
**is the prior for the mean used to model memory.**




**Table 2: Chronological control points and modelled ages for study sites in the GVWSA.**

| Site and Sample ID | Method | Material | Depth (cm) | ¹⁴C age (ybp) | ± | Mean probability age (cal yr BP) |
|---|---|---|---|---|---|---|
| **Frog Lake** | | | | | | |
| Core top | N/A | N/A | 0 | N/A | 10 | -62 |
| UOC-17575 | 14C | Bulk sediment | 35.5 - 36 | 879 | 16 | 760 |
| UOC-17531 | 14C | Leaves/ organics | 88 | 1337 | 20 | 1279 |
| UOC-17532 | 14C | Twig | 158 | 2492 | 20 | 2591 |
| UOC-17533 | 14C | Plant | 204 | 3267 | 21 | 3485 |
| UOC-17534 | 14C | Organic fragments | 236 | 3769 | 21 | 4133 |
| UOC-17535 | 14C | Needle | 342 | 5386 | 21 | 6213 |
| N/A | Tephra | Mazama ash | 395 | N/A | N/A | 7631* |
| UOC-17536 | 14C | Twig | 412 | 7298 | 23 | 8097 |
| UOC-17537 | 14C | Twig & Seed | 452 | 8671 | 24 | 9618 |
| UOC-17538 | 14C | Twig | 488 | 9573 | 25 | 10,907 |
| UOC-17539 | 14C | Needle | 548 | 11,256 | 30 | 13,138 |
| **Begbie Lake** | | | | | | |
| Core top | N/A | N/A | 0 | N/A | 10 | -59 |
| BEG-35 | 14C | Plant fragment | 29 | 280 | 20 | 382 |
| BEG-82 | 14C | Twig | 76 | 1435 | 20 | 1330 |
| BEG-110 | 14C | Plant fragment | 104 | 1875 | 25 | 1803 |
| BEG-174.5 | 14C | Twig | 168.5 | 3075 | 20 | 3236 |
| BEG-245 | 14C | Bark fragment | 239 | 3935 | 20 | 4398 |
| BEG-351 | 14C | Plant fragment | 345 | 5785 | 20 | 6570 |
| N/A | Tephra | Mazama ash | 406 | N/A | N/A | 7631* |
| BEG-477 | 14C | Twig | 471 | 8205 | 25 | 9153 |
| BEG-519.5 | 14C | Seed /plant fragment | 513.5 | 8885 | 25 | 10,047 |
| BEG-581 | 14C | Twig | 575 | 10,240 | 30 | 12,019 |
| BEG-627.5 | 14C | Needle and plant frag | 621.5 | 11,960 | 30 | 13,721 |
| **Swanson Lake** | | | | | | |
| Core top | N/A | N/A | 0 | N/A | 10 | -60 |
| UOC-21458 | 14C | Bulk sediment | 50- 51 | 1786 | 14 | 1679 |
| UOC-21459 | 14C | Bulk sediment | 115- 116 | 3146 | 15 | 3376 |
| UOC-17541 | 14C | Twig | 168 | 3861 | 21 | 4292 |
| UOC-17542 | 14C | Twig | 207 | 4404 | 21 | 4987 |
| UOC-17543 | 14C | Bark/cone | 278 | 5619 | 22 | 6387 |
| UOC-17544 | 14C | Twig | 326 | 6254 | 22 | 7164 |
| N/A | Tephra | Mazama ash | 366 | N/A | N/A | 7631* |
| UOC-17545 | 14C | Plant | 503 | 8983 | 25 | 10,162 |
| UOC-17546 | 14C | Plant | 551 | 9742 | 26 | 11,191 |
| UOC-17547 | 14C | Plant fragments | 591 | 10,549 | 26 | 12,513 |
| **Worley Lake** | | | | | | |
| Core top | N/A | N/A | 0 | N/A | 10 | -59 |
| UOC-245856 | 14C | Bulk sediment | 10 - 11 | 470 | 20 | 457 |
| UOC-24857 | 14C | Bulk sediment | 20 - 21 | 670 | 15 | 644 |
| UOC-24858 | 14C | Bulk sediment | 31 - 32 | 1110 | 20 | 1064 |
| UOC-17549 | 14C | Twig | 35 | 2738 | 20 | 1674** |
| UOC-21461 | 14C | Bulk sediment | 60- 61 | 3382 | 15 | 3593 |
| UOC-17550 | 14C | Needle | 89 | 3704 | 23 | 4011 |
| UOC-17551 | 14C | Plant | 122 | 3814 | 22 | 4395 |
| UOC-17552 | 14C | Plant fragments | 176 | 5214 | 22 | 5930 |
| UOC-17553 | 14C | Twig | 218 | 5387 | 22 | 6726 |
| N/A | Tephra | Mazama ash | 241 | N/A | N/A | 7631* |
| UOC-17554 | 14C | Plant | 260 | 7291 | 23 | 8104 |
| UOC-17555 | 14C | Twig | 304 | 8303 | 24 | 9,321 |
| UOC-17556 | 14C | Twig | 405 | 10,505 | 26 | 12,356 |

N/A: not available; *modelled age from Egan *et al*. (2015); **sample modelled outside the 95% confidence interval






## 3.2 Frog Lake

The early-Holocene Frog Lake pollen record (Fig. 3a), with ranges given in parenthesis, consists of *Pinus* (13-46%) and *Alnus* (29-48%) with *P. menziesii* (<0.5-23%) increasing after 10,000 cal yr BP. *Pteridium* (1.5-13%), *Rosaceae* (1.5-12%), *Salix* (1-9%), and *Poaceae* (<0.5-6%) were important understory constituents, together with *Apiaceae* (0-3%), *Artemesia* (0-2%) and
monolete ferns (0-4%). In the late-Holocene, *Cupressaceae* (0-22%), and *Tsuga heterophylla* (0-15%), along with *Picea* (0-8%) and *Abies* (0-5%), increase in abundance, while *Pinus* (5-15%) decreases. *Acer* (0-1%), *Quercus* (0-3%), *Caprifoliaceae* (0-6%), *Saxifragaceae* (0-1%) and Camassia (0-1%) emerge as additional distinguishing taxa. AP averages 83% in the early-Holocene, increasing to 85% in the late-Holocene (Fig. 3a). Comparing changes in fire-related plant functional groups between the early- and late-Holocene, invaders decrease from 71 to 42%, while avoiders and resisters increase from 5 to 23% and 9 to
16%, respectively.

Charcoal analysis was evaluated with a median SNI of 4.3 in the early-Holocene. While charcoal influx (CHAR) averages 0.5 *pieces cm$^{-2}$ year$^{-1}$*, morphotype influxes vary from 0.05, 0.12, and 0.06 *pieces cm$^{-2}$ year$^{-1}$* for graminoids, wood, and herbaceous/deciduous materials, respectively (Fig. 3b). Eleven fire events are identified, with peak magnitude averaging 43 *pieces cm$^{-2}$ peak*. FRIs range from 114-760 years with a mean FRI (mFRI) of 315 years. In contrast, median SNI was 3.0 in
the late-Holocene, with CHAR averaging 0.8-*pieces cm$^{-2}$ year$^{-1}$*. Mean morphotype influx is 0.05 (graminoids), 0.20 (wood), and 0.09 (herbaceous/ deciduous) *pieces cm$^{-2}$ year$^{-1}$*. Nine fire events were detected, with peak magnitude averaging 225 *pieces cm$^{-2}$ peak*. FRIs range from 38-646 years with a mFRI of 323 years (Fig. 7a).




**Figure 3: Early- and late-Holocene pollen and charcoal results for Frog Lake. (a) Pollen diagram from left to right: taxa with relative abundance >5% (black area represents scaled pollen abundance (%) with 5x exaggeration (grey curve), cumulative pollen abundance (%) for arboreal (AP) and non-arboreal pollen (NAP) types, and fire functional groups. (b) Charcoal diagram from left to right: charcoal concentration, CHAR with grey curve showing 95% threshold, charcoal morphotype influxes, significant positive fire-related peaks (black dots) and positive peaks with a low SNI (grey dots) at a 95% and 99% threshold, SNI, smoothed fire frequency, magnetic susceptibility with black representing raw values and grey 2X exaggeration.**



### 3.3 Begbie Lake

At Begbie Lake *Pinus* (12-40%), *Pseudotsuga* (9-43%), and *Alnus* (14-29%) dominate the tree component during the early-
Holocene, with *Pseudotsug*a peaking at ca. 9,600 cal yr BP. *Salix* (1-2%) is found in lesser amounts. The main shrub is
*Rosaceae* (1-4%), with *Pteridium* (4-12%), *Cyperaceae* (0.5-8%), and *Poaceae* (0-3%) also prevalent in the understory (Fig.
4a). In the late-Holocene, *Pinus* (5-24%) and *Pseudotsug*a (5-42%) continue to persist at the site as *T. heterophylla* (8-33%)
and *Cupressaceae* (4-17%) emerge as co-dominants. *Picea* (0-5%) and *Abies* (<0.5-5%) also increase. Slight decreases in
*Roasaceae* (0-2%), *Poaceae* (0-1.5%), and *Pteridium* (0-6%) are evident compared to the early-Holocene. An increase in
*Lysichiton* (0-2.5%) occurs after ca. 3000 cal yr BP while *Myrica* (0-10%) establishes vigorously at the site after ca. 1000 cal
yr BP (Fig. 4a). AP comprise an average of 80% of the pollen spectra in the early-Holocene versus 90% in the late- (Fig. 4a).
Invaders decrease in abundance between the early- (51%) and late-Holocene (29%) while avoiders (7%; 34%) and resisters
(20%; 25%) increase.

A median SNI of 3.5 was calculated from the early-Holocene charcoal analysis at Begbie Lake (Fig. 4b). CHAR averages 2.0
*pieces cm$^{-2}$ year$^{-1}$*. Eleven fire-related peaks are detected with peak magnitudes averaging 985 *pieces cm$^{-2}$peak*. FRIs range
from 88-704 years with a mFRI of 290 years (Fig. 7; a). In the late-Holocene, median SNI is 3.0. CHAR averages 1.4 *pieces
cm$^{-2}$ year$^{-1}$,* and eight positive peaks are identified along with average peak magnitudes of 91 *pieces cm$^{-2}$peak*. FRIs range from
176-726 years with a mFRI of 394 years (Fig. 7a).








**Figure 4: Early- and late-Holocene pollen and charcoal results for Begbie Lake. (a) Pollen diagram from left to right: taxa with relative abundance >5% (black area represents scaled pollen abundance (%) with 5x exaggeration grey curve), cumulative pollen abundance (%) for AP and NAP types, and fire functional groups. (b) Charcoal diagram from left to right: charcoal concentration, CHAR with grey curve showing 95% threshold, significant positive fire-related peaks (black dots) and positive peaks with a low SNI (grey dots) at a 95% and 99% threshold, SNI, smoothed fire frequency, magnetic susceptibility with black representing raw values and grey 2X exaggeration.**



## 3.4 Swanson Lake

The early-Holocene tree stratum at Swanson Lake (Fig. 5a) contains *Pinus* (28-62%), *Pseudotsuga* (4-12%), and *Alnus (0-50%)*. *T. heterophylla* (1-13%) increases throughout the interval, with *Cupressaceae* (<0.5-2%), *Abies* (<0.5-3%) and *Salix* (0-2%) contributing to background forest composition. The understory consists of *Rosaceae* (0-4%), with *Poaceae* (0-2%), *Plantaginaceae* (0-2%), and *Pteridium* (2-6%). In the late-Holocene, *T. heterophylla* (8-34%) increases along with *Cupressaceae* (1-9%), *Picea* (0-5%) and *Abies* (0-5%). *Pinus* (5-39%), *Pseudotsuga* (1.5-12%), and *Alnus* (21-46%) remained relatively constant. In general, shrubs and herbs decrease, except for a minor expansion of *Ericaceae* (0-2%). AP types average 90% of the population in the early-Holocene as opposed to 95% in the late-Holocene (Fig. 5a). Invaders decrease from an average of 74% in the early-Holocene to 61% in the late-Holocene, while avoiders increase from 7% to 27%. Resisters decrease at the site from 8% in the early-Holocene to 6% in the late-Holocene.

Evaluation of the early-Holocene charcoal analysis produces a median SNI of 3.2 (Fig. 5b). CHAR averages 0.8 *pieces cm$^{-2}$ year$^{-1}$* while charcoal influx per morphotypes is 0.12 (graminoids), 0.19 (wood), and 0.15 (herbaceous/ deciduous) *pieces cm$^{-2}$ year$^{-1}$*. Seventeen positive fire events are identified with peak magnitudes averaging 20 *pieces cm$^{-2}$ peak*. FRIs range from 57-361 years with a mFRI of 195 years (Fig. 7a). During the late-Holocene, analysis results in a median SNI of 2.4. CHAR averages 0.4 *pieces cm$^{-2}$ year$^{-1}$* and morphotype influx is 0.02, 0.12, and 0.03 *pieces cm$^{-2}$ year$^{-1}$* for graminoid, wood and herbaceous/ deciduous charcoal, respectively. Ten positive peaks are recorded with peak magnitudes averaging 15 *pieces cm$^{-2}$ peak*. FRIs range from 80-620 years with mFRI of 311 years.





**Figure 5: Early- and late-Holocene pollen and charcoal results for Swanson Lake. (a) Pollen diagram from left to right: taxa with relative abundance >5% (black area represents scaled pollen abundance (%) with 5x exaggeration grey curve), cumulative pollen abundance (%) for AP and NAP types, and fire functional groups. (b) Charcoal diagram from left to right: charcoal concentration, CHAR with grey curve showing 95% threshold, charcoal morphotype influxes, significant positive fire-related peaks (black dots) and positive peaks with a low SNI (grey dots) at a 95% and 99% threshold, SNI, smoothed fire frequency, magnetic susceptibility with black representing raw values and grey 2X exaggeration.**





### 3.5 Worley Lake

At Worley Lake *Pinus* (7-29%), *Pseudotsuga* (3-20%), and *Alnus* (32-61%) are the main tree species in the early-Holocene
(Fig.6a). *T. heterophylla* (0-22%) increases at the top of the interval while *Salix* (0-1%) is present in detectable quantities until ca. 8000 cal yr BP. *Roasaceae* (0-2.5%) helps define the shrub layer. *Poaceae* (0-1%), *Cyperaceae* (0-2%), monolete ferns (0.5-3%), and *Pteridium* (1-9%) make up the understory. In the late-Holocene, *T. heterophylla* (10-40%) and *Cupressaceae* (3-35%) dominate the overstory, alongside *Pinus* (8-20%). The cuppressaceous pollen, possibly derived from both *T. plicata* and *C. nootkatensis*, increases considerably over the last 1000 years, during which time *Pseudotsuga* (1-5%) pollen decreases.
*Picea* (0.5-5%) and *Abies* (1-4%) form minor components of the forest. *Alnus* (13-44%) decreases slightly compared to the early portion of the record, as do most understory taxa. AP types make up 89% of the record in the early-Holocene, contrasted with an average of 93% in the late-Holocene (Fig. 6a). Fire-related functional groups show invaders decreasing from an average of 63% in the early-Holocene to 45% in the late-Holocene as avoiders increase from 10% to 46%. Resisters also decrease from 9% in the early-Holocene to 3% in the late-Holocene.

Evaluation of charcoal analysis for the early-Holocene record produce a median SNI of 4.9 (Fig. 6b). CHAR averages 1.0 *pieces cm⁻² year⁻¹* while influx for charcoal morphotypes averages 0.09 (graminoids), 0.26 (wood), 0.14 (herbaceous/ deciduous) *pieces cm⁻² year⁻¹*. Thirteen positive fire-related peaks are documented with peak magnitudes averaging 227 *pieces cm⁻² peak.* FRIs range from 87-377 years and mFRI is 230 years. In the late-Holocene, median SNI is 3.5 and CHAR averages 0.25 *pieces cm⁻² year⁻¹*. Influx for charcoal morphotypes averages 0.01 (graminoids), 0.08 (wood), and 0.03 herbaceous/
deciduous) *pieces cm⁻² year⁻¹*. Four significant peaks were detected with peak magnitudes averaging 251 *pieces cm⁻² peak*. FRIs range from 609-1856 years and mFRI was 1082 years.





**Figure 6: Early- and late-Holocene pollen and charcoal results for Worley Lake. (a) Pollen diagram from left to right: taxa with relative abundance >5% (black area represents scaled pollen abundance (%) with 5x exaggeration grey curve), cumulative pollen abundance (%) for AP and NAP types, and fire functional groups. (b) Charcoal diagram from left to right: charcoal concentration, CHAR with grey curve showing 95% threshold, charcoal morphotype influxes, significant positive fire-related peaks (black dots) and positive peaks with a low SNI (grey dots) at a 95% and 99% threshold, SNI, smoothed fire frequency, magnetic susceptibility with black representing raw values and grey 2X exaggeration.**



### 3.6 Comparison of FRIs, Biomass Burning, and Climate

A Kruskall-Wallis test ($H$=16, $p$=0.025) indicates that FRIs differ either temporally or spatially in at least one group. Subsequent Wilcoxon test ($W$=0, p.adjusted=0.04, $r$=0.68) on paired dependent samples shows that FRIs change significantly from the early- to late-Holocene at Worley Lake, but not at other sites. A Dunn's pairwise comparison ($Z$=3.35, p.adjusted (Bonferroni)= 0.022) of all independent groups (i.e. excluding comparisons between early- and late-Holocene FRIs from the same site) shows notable differences in FRIs between Worley Lake in the late-Holocene and Swanson Lake in the early-Holocene but did not detect differences between other pairings (Fig. 7a).

Transformed and binned $Z$-scores of charcoal influx are relatively synchronous during the early-Holocene, based on agreement between site-specific locally regressed curves and narrower boot-strapped confidence intervals (CI). Charcoal influx is highest between ca. 9500-11,000 cal yr BP (overall above average $Z$-scores), decreasing thereafter (Fig. 7b). The curve for Frog Lake corresponds to a minimum for the early-Holocene, falling outside of bootstrapped 95% confidence intervals. The late-Holocene shows greater variability, with charcoal influx between ca. 2000-3000 and 0-500 cal yr BP being somewhat synchronous. However, a pattern of asynchroneity is also evident at this time at Frog and Worley lakes, with dips in biomass burning at Worley Lake coinciding with increased burning at Frog Lake.

Estimated MAP from DWHI is 1015 mm at Frog, 965 mm at Begbie, 1255 mm at Swanson, and 1340 mm at Worley lakes during the early-Holocene (Fig. 8a). In contrast, precipitation increases in the late-Holocene to 1265, 1610, 2550, and 3170 mm, respectively, with the wettest climate prevailing at Worley Lake through time. Extraction of MAP (BIO12) from the PALEO-PGEM and HadCM3 series are consistent with the trend that precipitation increased from early- (PALEO-PGEM ranges = 1150-1240 mm; HadCM3 =1444-1588 mm) to late-Holocene (PALEO-PGEM = 1250- 1254 mm; HadCM3 = 1572 - 1627) in the region (Fig. 8b). Further, PALEO-PGEM and HadCM3 (BIO5) reveals that Tmax was relatively high in the early-Holocene (PALEO-PGEM = 17.3-18.5 °C; HadCM3 = 24.6-25.3 °C), peaking between 8000-10,000 cal yr BP (Fig. 8b). Thereafter, temperature steadily decreases through the late-Holocene (PALEO-PGEM = 17.0-16.2 °C; HadCM3 = 22.4-23.2 °C). Chironomid inferred MJAT likewise establishes warmer conditions in the early-Holocene (11.1-13.9 °C) compared to late-Holocene (5.7-8.8 °C), while the MJAT anomaly also suggests that temperatures were consistently 0.5-2°C warmer than modern in the early-Holocene (Fig. 8b.; Pellatt *et al*., 2000; Lemmen and Lacourse, 2018).





**Figure 7: Comparison of FRIs and biomass burning. (a) FRI box and whisker plots comparing sites. Boxes represent the first Q1 and third Q3 quartiles, black line shows the median, whiskers are minimum and maximum range and black dots are outliers. Top black bracket and asterisks shows results of Wilcoxon signed rank test on paired samples (W=0, ρ.adjusted=0.04, r=0.68) and results of Dunn's pairwise comparison (Z=3.35, ρ.adjusted (Bonferroni)= 0.022). (b) Standardized charcoal influx (Z scores) showing biomass burning based on a regression of presampled and binned records at 20-year time step and 500-year half width smoothing. Bootstrapped confidence intervals (CI) and mean from 5000 resampled iterations. Coloured curves present local fits for each site.**




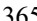

**Figure 8: Regional climate reconstruction for the Holocene. (a) Boxplots for Douglas-fir – western hemlock index estimates of mean annual precipitation MAP (DWHI). Blue dots are means. (b) From left to right: MAP (BIO12) from PALEO-PGEM series (Baretto *et al*., 2023), MAP (BIO12) from HadCM3 (Beyer *et al*., 2020), down core DWHI precipitation estimates for each site, mean maximum**

**monthly temperature (Tmax) in degrees Celsius from PALEO-PGEM series (Baretto *et al*., 2023), Tmax from HadCM3 (Beyer *et al*., 2020), chironomid inferred mean July air temperature (MJAT) (Pelatt *et al*., 2000), and chironomid inferred (MJAT) anomalies (Lemmen and Lacourse, 2018).**





## 4 Discussion

### 4.1 Early-Holocene Climate, Vegetation, and Fire

In the early-Holocene, climate reconstructions for western North America identify a warm-dry period with greater seasonality and augmented summer drought due to increased summer insolation (Mathewes and Heusser, 1981; COHMAP, 1988; Hebda, 1995; Rosenberg *et al.,* 2004; Brown *et al.*, 2006). Multi-proxy paleoclimate data from sites in BC and spatially downscaled global climate series simulations suggest that regional temperature peaked around ca. 9000 – 10,500 cal yr BP (Walker and Mathewes, 1987; Pellatt *et al.*, 2000; Lemmen and Lacourse, 2018; Beyer *et al.*, 2020; Brown *et al.*, 2022; Baretto *et al.*, 2023;

Fig. 8b). Pollen derived DWHI values from the lakes within the GVWSA similarly suggest widespread dry conditions during the early-Holocene (Brown *et al.*, 2019), with MAP <1500 mm. For perspective, these conditions are consistent with the driest sites in the region today and in the range of the modern and relatively dry Coastal Douglas-fir (CDF) zone (Meidinger and Pojar, 1991; Wong *et al.*, 2004; Fig. 1b, c; Fig. 8a).

In response to a warming climate in the early-Holocene, open, xeric forests largely prevailed across the GVWSA, with NAP

averaging 10-20% amongst sites (Figs. 3a, 4a, 5a, 6a). Overall, the canopy was dominated by shade intolerant, thick-barked, *P. menziesii*, a fire resister that can survive frequent low- to moderate-severity fires (Agee, 1993; Wirth, 2005). Except for *Pinus*, other coniferous taxa were scarce in the area until after ca. 9000 cal yr BP, when *Picea*, *Abies*, *T. heterophylla* and *Cupressaceae* gradually increased in abundance. Post-fire invaders, such as *Pinus* (most likely *Pinus contorta*), *Alnus, Salix,* and *Poaceae* are also abundant, suggesting regular disturbances that favoured pioneering species with high rates of dispersal,

germination, and/or re-sprouting (Agee, 1993; Wirth, 2005). *Pteridium,* a fire promoting species whose deep rhizomes enable it to endure disturbance, is also pervasive in the early-Holocene (Crane, 1990).

At Frog Lake, in the driest eastern regions of the GVWSA, *Poaceae* pollen peaks at 6%, possibly reflecting the presence of woodland-like communities or a patchwork of meadows in a forested system (Allen *et al.,* 1999). Open woodland and shrubland consisting of *Salix*, *Rosaceae,* and *Apicaeae* thrived and were probably common (Fig. 3a). *Artemesia* also appears

at this time, along with diagnostic amounts of *Selaginella* that are indicative of open rocky bluffs, likely prevailing on southwestern aspects above and below the study site (Allen, 1999; Fig. 1g). Begbie Lake is mostly characterized by its surrounding high proportion of *P. menziesii*, which peaked at ca. 9500 cal yr BP. *Poaceae* and *Rosaceae* appear in lesser amounts than at Frog, but their presence, along with a strong signal for *Pteridium* support the theory that open forests with well-developed understories occupied eastern areas (Brown *et al.*, 2019; Fig. 4a). The incursion of *Cyperaceae* at the site near

the top of the interval could indicate the beginning of regional moistening and resultant wetland development, which currently characterises the low-lying terrain around Begbie.

In the west, *Pinus* dominates at Swanson Lake in the early-Holocene, an indication that vegetation on the exposed ridge was frequently disturbed (Fig. 5a). Distinguishing quantities of *Plantaginaceae* pollen (believed to be *Penstemon* spp.) in the early-




Holocene reflect dry rocky out-crops around the site (Douglas *et al*., 1999). Worley Lake is characterised by an increase in *P.*
*menziesii* at ca. 10,000 cal yr BP, along with abundant *Alnus* and perceptible *Roasaceae* and *Poaceae* (Fig. 6a). *T. heterophylla*
increases sharply after ca. 9000 cal yr BP, signaling that climate driven changes in forest composition and structure may have
been occurring earlier in the western part of the study region. Both Swanson and Worley Lakes had otherwise underdeveloped
shrub and herbaceous layers, implying that these western forests were less open than contemporaneous eastern forests. Even
so, high relative proportions of *P. menziesii*, *Poaceae*, *Rosaceae,* and *Pteridium* do not indicate a closed canopy at either site
(Fig. 5a, 6a). Other pollen records from southern Vancouver Island establish that CDF-like forests in the early-Holocene could
have extended 100 km west of the zone's current range (Brown and Hebda, 2002a), with the strong presence of *P. menziesii*
at Swanson and Worley helping verify the timing and extent of that expansion.

Regarding early-Holocene fire disturbance, median SNI exceeded the acceptable threshold (>3; Kelly *et al*., 2011) at all sites,
demonstrating that the records were suitable for peak detection. At Swanson Lake, however, median SNI (3.2) was slightly
lower compared to the other sites despite comparable sediment accumulation rates and good sampling resolution. Noting that
Swanson is the shallowest of the lakes studied, it is posited that it experienced more charcoal mixing, though not enough to
entirely impede peak detection. The charcoal records reveal that fires were indeed more frequent at all sites across the water
supply area when compared to the late-Holocene interval, with mean early-Holocene FRIs ranging from 195 – 315 years (Fig.
7a). Critically, an east-west delimitation of fire regimes is not evident in the early-Holocene. Lower peak magnitudes at Frog
and Swanson lakes compared to Begbie and Worley lakes suggests that less severe events characterised the former locations,
which share similar site characteristics including exposed locations and rocky out-croppings. The peak magnitudes at Begbie
and Worley lakes demonstrate that large or high-severity fires occurred where fuels were available. Based on charcoal
morphotype distinctions, the proportion of graminoid-to-woody charcoal influx was higher at all sites in the early-Holocene,
with Frog and Swanson capturing slightly higher ratios of graminoid burning (Table 3). This implies that both surface and
crown fires were occurring with surface fires potentially more frequent at open, dry, and/or exposed sites. Indeed, several
charcoal magnetic peaks coincide with charcoal peaks, indicating the presence of high-severity fires (Rummery *et al*., 1981;
Gedye *et al.,* 2000; Dunnette *et al.,* 2014; Figs. 3b, 4b, 5b, 6b).

**Table 3. Fuel Morphotype Index (FMI) for sites in the GVWSA. Values closer to 1.0 indicate higher rates of wood combustion. Begbie Lake not included as morphotypes were not analysed at the time the core was sampled.**

|  | Frog Lake | Swanson Lake | Worley Lake |
|---|---|---|---|
| *FMI* early-Holocene | 0.72 | 0.65 | 0.75 |
| *FMI* late-Holocene | 0.82 | 0.87 | 0.84 |

Transformed and amalgamated total charcoal influx from each site shows that overall biomass burning was synchronous and
widespread in the early-Holocene. The highest rates of charcoal influx occurred around ca. 10,500 cal yr BP, coinciding with
the regional thermal maximum, and decreased steadily after ca. 9500 cal yr BP (Mathewes and Heusser, 1981; Walker and





Pellatt, 2003; Fig. 7b), with a overall reduction at 8200 cal yr BP that could be related to northern hemisphere cooling (Alley
435    *et al.,* 1997; Barber *et al*., 1999). In the driest area, around Frog Lake, lower rates of charcoal influx could reflect a moderately
fuel constrained landscape (Fig. 7b).  Overall, the pollen and charcoal records reveal that CDF-like temperate forests expanded
westward in the GVWSA during the warm-dry early-Holocene, supporting an active fire regime with surface and crown
disturbance (Gavin *et al.,* 2003a; Giuliano and Lacourse, 2023; Brown et al., 2019, 2022; Hebda *et al*., 2024).

## 4.2 Change in Late-Holocene Climate and Shifting Fire Regime

440    Cooling, moistening, and the establishment of contemporary precipitation patterns on southern Vancouver Island occur after
6000 cal yr BP (COHMAP, 1988; Hebda, 1995; Brown *et al*., 2006; Rosenberg *et al*., 2004). The decrease in temperature is
variously related to decreasing solar activity from orbital forcing and changes in oceanic currents (Wanner *et al*., 2008; Marlon
*et al*., 2012). Regional temperature reconstructions show that the monthly mean maximum temperatures were declining in the
late-Holocene (Mathewes and Heusser, 1981; Pellatt *et al.,* 2000; Beyer, 2020; Baretto *et al*., 2023; Fig. 8b), with substantial
variability in MJAT anomalies (Lemmen and Lacourse, 2018). Regarding precipitation, a stronger east-west gradient
characterised the GVWSA compared to the early-Holocene. For example, while MAP did not change profoundly at eastern
Frog Lake through time, a marked early-to-late Holocene shift is evident at western Worley Lake, with precipitation increasing
from 1340 mm in the early-Holocene to >3000 mm in the late-Holocene (Fig. 8a).

The changes in climate had a noticeable effect on vegetation composition and distribution. Arboreal pollen increased in
abundance across all sites as *T. heterophylla* and *Cupressaceae* (*T. plicata*) expanded throughout the region (Fig 4a, 5a, 6a,
7a). Notably, both species are shade tolerant, late-successional fire avoiders that are poorly adapted to fire disturbance. *Picea*
(likely *Picea stichensis*) and *Abies* (likely *Abies amabilis*)*,* also avoiders, likewise increased in abundance. Fire invaders
decreased at all sites in the late-Holocene, suggesting that mixed-conifer late-seral closed forests with infrequent disturbance
developed across much of the GVWSA.

Despite these significant changes, fire-adapted *P. menziesii*, remained co-dominant at Frog and Begbie lakes in the east. At
Frog Lake, a signal for *Poaceae* persisted even with the expansion of *T. plicata* and *T. heterophylla* (Fig. 3a). Along with
sustained understory indicators such as *Roasaceae* and *Pteridium,* it is likely that eastern forests in the GVWSA maintained
an open component. Between ca. 2000 – 3000 cal yr BP, *Caprifoliaceae* (possibly *Lonicera hispidula*) emerged at Frog Lake
in the understory. *L. hispidula*, or pink honeysuckle, is typically found in mesic to dry forests where it is sometimes associated
with oak (Pojar et al.,1994; Fairbarns, 2023). At Begbie Lake, a decrease in *Pteridium,* in conjunction with the emergence of
*Lysichiton*, *Myrica,* and further persistence of *Cyperaceae* tracks expanding wetlands around the site due to increased moisture
input (Brown *et al*., 2019; Fig. 4a).

*Quercus (*likely *Quecus garryana*) also emerged as a minor forest constituent around Frog and Begbie lakes (and to a lesser
degree Swanson and Worley lakes) between ca. 2000 – 3000 cal yr BP (Fig. 3a, 4a, 5a, 6a). *Q. garryana* is classified as a fire



endurer with the ability to re-sprout from lateral roots or grow new crowns after fire (Agee, 1996). Its expansion during the mid-Holocene is well documented from sites on east Vancouver Island (Pellatt *et al*., 2001; Lucas and Lacourse, 2013), whereas the late-Holocene persistence of *Q. garryana* in interior island forests is less recognized (Allen, 1995; Brown *et al*., 2019). Together with the species' response to fire, it has been linked to Indigenous land management in the region (Pellatt and Gedalof, 2014; Pellatt *et al*., 2015; Brown *et al*., 2002b, 2019; Barlow *et al*., 2021). An archaeological impact assessment for
the CRD at the Sooke Lake Reservoir found artefacts of Marpole origin, a cultural period that dates to as early as 2350 yrs BP and overlaps with the *Q. garryana* signal in the watershed (Burley, 1980; Vincent *et al*., 2002), suggesting that the oak signal may be related to human activity in the eastern sector of the GVWSA at that time.

In the west, *Pinus* and *Alnus* remained dominant at Swanson Lake reflecting the sites' continued exposure to disturbance. *T. heterophylla* increases steadily through the interval along with *Picea* and *Abies* while *P. menziesii* remains a constant.
Traceable amounts of *Ericaceae* (*Vaccinium* spp.) pollen appear after ca. 2000 cal yr BP (Fig. 5a). Most herbaceous taxa fade from the record. At Worley Lake, a substantial decrease in *P. menziesii* coincides with increases in *T. heterophylla*, *Picea*, *Abies,* and *Cupressaceae* (Figs. 5a, 6a).

Regarding late-Holocene fire disturbance, a marked decrease in SNI occurred across all sites, with values hovering around three at most sites. It is possible that this decrease is related to local wetland expansion intercepting overland charcoal inputs
(Agee, 1993; Higuera *et al*., 2007; Kelly *et al*., 2011). The SNI at Swanson Lake is of greatest concern since it is generally below the threshold for suitable peak detection (Fig. 5b). However, the preponderance of fire-invading *Pinus* around Swanson Lake suggests that fires did indeed continue to burn around the site, with the emergent estimates of FRIs consistent with those from Begbie and Frog lakes. Thus, while cautious interpretation of the late-Holocene fire record is warranted for Swanson Lake, it is presumed that the results generally reflect the fire regime.

Overall, fire is less frequent in the GVWSA during the late-Holocene compared to the early-Holocene. Mean FRIs range from 311 – 1082 years, reflecting slightly longer times between fires (Fig. 7a). While fire severity varies between sites, all basins record a period when fires were likely small and/or of low severity between ca. 1500 and 3000 cal yr BP, as evidenced by generally low peak magnitudes and magnetic susceptibilities. A marked increase in magnetics occurs at the top of all records, likely tracking human expansion and logging development (Wu *et al*., 2015; Magiera *et al*., 2019; Figs. 3b, 4b, 5b, 6b).
Additionally, larger peak magnitudes, coinciding with increases in magnetic susceptibility, are noted at Frog Lake after 1500 cal yr BP and at Worley Lake at 700 cal yr BP. Elevated inputs of woody charcoal morphotypes correspond with these events, implying that these fires were of higher severity and likely initiated crown involvement or consumed coarser fuels (Fig. 3b, 6b). Likewise, the *FMI* showed increases in the combustion of woody materials at all sites in the late-Holocene (Table 3).

In general, transformed charcoal influx showed that biomass burning exhibited greater variability across the water supply in
the late-Holocene compared to the early-Holocene (Fig. 7b). While periods of synchronous biomass burning occurred,





asynchonity also emerged. For example, at Worley Lake, charcoal influx was much lower compared to the other sites and fluctuated considerably through time, possibly tracking background charcoal as climate became wetter and more variable. As canopy closed at the site, it likely intercepted precipitation reducing residual charcoal influx in the absence of local fires. Although there appears to be an overall decrease in charcoal influx in the GVWSA during the late-Holocene, curves show that

some increases did occur as climate cooled and moistened. These trends have been identified in other regional records (Brown and Hebda, 2002b; Hallett *et al*., 2003), with several causal mechanisms proposed to explain the pattern. For example, while open forests largely prevailed in the early-Holocene, denser forests with elevated fuel loads expanded in the late-Holocene, potentially contributing to a contrasting mosaic of stand replacing events (Gavin *et al*., 2007). Increased lightning with greater climatic variability could also promote fires given higher ignition probability during periods of seasonal drought. On the

adjacent mainland, Hallet *et al*., (2003) reveal elevated fire activity from ca. 1500 - 2500 cal yr BP, referring to it as the Fraser Valley Fire Period. In the GVWSA, coincident increases in burning around 1000 cal yr BP may better align with the Medieval Climate Anomaly (ca. 750-1250 cal yr BP). Lastly, the possible escalation of human activity has been proposed along with cultural uses of fire (Brown and Hebda, 2002b; Gavin *et al*., 2007; Walsh *et al.,* 2015). Findings from both the pollen and charcoal records in the late-Holocene correspond well with the establishment of modern moist CWH forests, which experience

less frequent but mixed-severity fire. These fire regimes are complex and sometimes drivers of disturbance are not clear, but they can be characterised by periods of fire absence that often leads to larger stand-replacing events (Agee, 1995; Brown *et al*., 2002a; Gavin *et al*., 2007, Brown *et al*., 2019).

**4.3 Informing Future Fire Regime Change in Temperate Forest**

The potential for past trends to reflect future changes in climate, vegetation, and fire is conceivable given that palaeoecological

interpretations and predictive models are in good agreement. For example, the substantial changes in fire disturbance that occur at Worley Lake demonstrate the extent of shifting fire regimes in the past, providing first-order insights into possible future conditions (Figs. 6b, 7a). Early-to-late Holocene differences in DWHI establish a historical precedent for a return to a much drier climate (Fig. 8a). While forecasting precipitation is complicated by seasonal variability and low confidence (Bush and Lemmen, 2019), predictions for the region suggest overall increases in annual precipitation, coupled with worsening summer

drought. Coastal regions could potentially see a 23% (-48 to -5%) reduction in summer precipitation in a high-emissions scenario (SSP5-8.5) compared to baseline (Bush and Lemmen, 2019; PCIC, 2024). Mean annual temperature in British Columbia is projected to increase by 1.6 (1.1, 2.1) – 5.2 (4.3, 6.5) °C by the end of the century, depending on the emissions scenario (RCP2.6, RCP8.5) and region, with the coast experiencing the least amount of warming (Bush and Lemmen, 2019; PCIC, 2024). These changes are in line with the estimates of past mean monthly maximum temperature fluctuations (2 – 5 °C)

during the early-Holocene but are projected to occur on a much shorter time span.

Forest change simulations align well with observations of past vegetation change. For example, the increase in *P. menziesii* that previously occurred at Worley Lake in the early-Holocene may recur in the future based on models examining climate



niche envelopes (Wang *et al*., 2012). The likelihood of these predictions is reinforced by the pollen record, which establishes the existence of a past *P. menziesii*-dominated forest at a site that is currently well-situated in CWHmm (Figs. 1b, 6a). Although

Wang *et al*., (2012) found that the area suitable for *P. menziesii* could double by the end of the century, CDF forest is only expected to expand with a net gain that reflects a 20% increase in its currently restricted range (0.2 million ha). The more nuanced interpretation is *P. menziesii* increases will be affiliated with the likely expansion of drier CWHxm as it displaces Vancouver Island interior and western rainforest. Increased disturbance and preferential reforestation with climate adapted species will also favor future expansion of *P. menziesii* (Uchytil, 1991).

With respect to fire disturbance, the paleofire reconstruction similarly favours model forecasts. Current trends suggest that wildfire disturbance is increasing in the region (Hanes *et al*., 2019; Parisien *et al*., 2023). Characterization of the early-Holocene fire regime in this study proposes that fire increased regionally in the past (Pellatt *et al*., 2001; Lucas and Lacourse, 2013; Cwynar, 1987; Allen, 1995; Brown and Hebda, 2002a; Fig. 6c). At Frog Lake, the most xeric site, FRI's change little between the early- and late-Holocene despite the former being warmer and drier (Fig. 7a). Given the similarities in FRIs

between the early- and late-Holocene, the lower charcoal flux may indicate that the fire mosaic was generally less severe in the early-Holocene, likely in response to reduced fuel loading. Further, smaller peak magnitude, in conjunction with forests dominated by *P. menziesii*, exemplifies a fire-resistant type of community. During the late-Holocene, a period of similar fire regime characteristics prevails at the site until ca. 2000 cal yr BP. Archaeological evidence and oral histories cite hunting as one of the likely cultural uses for the area around the Sooke reservoir (Eldridge and Seip, 2002). Along with the presence of

*Q. garryana* during the late-Holocene at Frog and Begbie Lakes, it is feasible that fire was used to maintain eastern forests for that purpose. Given these interpretations, a future fire regime under a warmer, drier climate might look like one that supports frequent low-severity fires in the eastern regions of the GVWSA as fuel loads become lighter. However, paleofire reconstructions also show that larger magnitude events are possible in the interim, as observed at Frog Lake between ca. 500 – 1500 cal yr BP. During the 2024 wildfire season, the Old Man Lake fire burned 230 ha just 5 km south of the Sooke Reservoir

and 8 km west of Frog Lake within CWHxm. Human causes were suspected, although it was above average July temperatures and persistent drought conditions on the coast that subsequently elevated fire danger ratings and led to the fires' spread.

Swanson similarly experienced high rates of fire disturbance in the past, but comparison of FRIs is complicated by the low SNI during the late-Holocene. Swanson Lake is located on a ridge that has been identified as an area of high-to-extreme wildfire risk in recent models, driven in part by the probability of ignition from lightning (Perakis *et al*., 2018). Two fires

ignited from lightning near the study site during the 2020 season. These fires were kept small in size (5.9 and 2.1 ha) due to aggressive suppression, though they illustrate the fire potential (Scott, 2020; BC Wildfire Service, 2024). The site's location and exposure to elevated ignition probability, could be a sustained driver of high-frequency, low-severity fires through both the early and late-Holocene intervals.



The significant difference in early- versus late-Holocene FRIs, coupled with changes in charcoal influx and the prevalence of high peak magnitude events at Worley Lake during the early-Holocene, indicate that the site was previously more vulnerable to fire than it is today (Fig. 6b, 7a,b). Dye *et al*., (2023) found that the largest magnitude of projected changes in burn probability, fire size, and number of fires corresponds with the coolest, wettest, and most northern areas in their simulations of future fire regimes for coastal temperate forests of the Pacific Northwest, reflecting settings similar to those around Worley Lake. The current vulnerability of the region to large stand-replacing events as fire regime transitions appears imminent and it is unlikely that cooler-moister forests in the west will be spared.

## 5 Conclusion

Comparison of climate, vegetation, and fire at four sites along ecological and climate gradients in a BC coastal temperate forest, revealed a significant temporal shift in fire regime between the early- and late-Holocene intervals. The magnitude of change was most pronounced at the coolest and wettest end of the study area, suggesting that top-down climatic forcing was a significant driver of fire in the past, even in wetter forests that are less prone to disturbance today. As first-order analogues, the records suggest that more fire will be present on the land in the future, a finding that is consistent with predictive models that forecast increased fire associated with climate change and vegetation response (Wang *et al.,* 2012; Wang *et al*., 2015, 2017; Sheehan *et al*., 2015; Dye *et al*., 2023). Yet, there appears to be an indication that on both ends of the gradients, intrinsic factors play an important role in fire regime variability; namely, feedback from vegetation response to climate and fire appears to exert bottom-up control over fire regimes due to changes in the structure of forests and fuels. In some cases, there is a coincidence of more open *P. menziesii* pollen assemblages with a charcoal record that insinuates periods of lower fire-severity. Where late-seral closed forests emerge in the late-Holocene pollen sequence, stretches of fire absence in the charcoal record are punctuated by large or high-severity events.

These differences have strong implications for management in a high-value water supply area. For instance, the decision to suppress a wildfire is one that demands careful evaluation of costs (both financial and human-life) and impacts (downstream effects of the disturbance, especially on water quality). One consideration is that exclusion and suppression can result in negative consequences for ecosystems (Hoffman *et al.,* 2019; 2021). For example, wildfires can provide valuable ecosystem services, such as pest regulation and nutrient cycling (DeLuca *et al*., 2016; Pausas and Keeley, 2019). Although a consensus has not yet emerged, it is clear that under the right conditions, negative feedbacks from suppression can add to a build-up of fuels and alter fire regimes, leading to more impactful high-severity fires (Stephens *et al*., 2014; Hai *et al*., 2023).  All this must be weighed against the overarching objective to protect an essential resource. Indeed, by showing how changes in past climate altered vegetation and fire regime, this study provides critical insight into the current and future susceptibility of managed high-value watersheds to anthropogenic climate change impacts.

**Data availability**

Data are available on the Neotoma Database. DOI are still being processed.

**Author contributions**

CB, JAT, and KJB conceptualized the research. DRH, KJB, and CB conducted fieldwork and core collection. DRH and KJB completed magnetic susceptibility profile of sediment cores. DRH sampled cores, and processed charcoal. NC processed pollen. DRH completed pollen and charcoal identification and tabulation for Worley, Swanson, and Frog Lakes. NC completed

pollen identification and tabulation for the Swanson record. DRH performed analysis and prepared figures. DRH and KJB wrote the paper with input from all authors.

**Competing interests**

The corresponding author has declared that none of the authors has any competing interests.

**Acknowledgements**

We would like to thank Henry Hart for his contribution to surveying and characterizing prospective study sites within the Greater Victoria Water Supply Area. Thank you Nicholas Hebda for prior work completed at Begbie Lake and for use of the data from the Begbie core for this study. We would also like to recognize Joel Ussery and the CRD for their support and access to the watershed and their commitment to science and research in this protected area.

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
