# Peer review of "Contrasting early- and late-Holocene vegetation and wildfire regimes in a high-value drinking water supply area, Canada"

_EGUsphere, 2025_

## Referee Comment (RC1)

Horrelt et al. "Contrasting early- and late-Holocene vegetation and wildfire regimes in a high-value drinking water supply area, Canada"

**General comments:**

This is an excellent, high-quality study. It provides a meaningful and timely contribution to paleoecology and paleofire research. The study is reasonably designed, with a rigorously applied methodology, and the data are extensive and convincing. Its main strength is its strong synthesis of multiple proxies (pollen, charcoal, and climate reconstructions) across a clear ecological gradient, addressing key questions about past and future fire regimes. The manuscript aligns with the scope of *Climate of the Past* by using a paleo perspective to deepen our understanding of current and future climate change impacts.

The manuscript presents excellent research that could be further enhanced by refining its storytelling. I recommend clarifying the introduction to clearly state the knowledge gap, using interpretive topic sentences to frame the results as a cohesive story, and starting discussion sections with a summary of key findings. To improve clarity, suggest synthesizing site details by emphasizing the east-west gradient and moving minor taxonomic data to an appendix. A final review to ensure concise, active language and a stronger conclusion that clearly connects the fire regimes to management implications would further increase the paper's impact.

**Specific comments:**

**[Introduction]**

The authors introduce the early-Holocene as a potential analogue but correctly caution that "the past does not provide a direct analogue." However, you can strengthen the connection by explicitly clarifying that, although not a direct analogue, the early-Holocene represents a climate state that may resemble near-future projections, making it an important period to study for understanding processes and sensitivities.

In the final objective paragraph, the phrase "to determine if past conditions could reflect evolving trends" (P2. L63) is somewhat vague. Do the authors mean "...to evaluate if past fire regimes provide insights into contemporary fire activity"?

- P2. L38. "Fishcer et al., 2015" is not in the Reference list.
- P2. L48. "None the less" to "Nonetheless"
- P2. L53. "imply" to "implies"
- P2. L57. " changing fire regime " to "a changing fire regime" or "changed fire regimes"

**[Study setting]**

This section is very detailed, but the authors could clarify the focus by separating the basic context from the methodological details. To strengthen the introduction, it should emphasize the rationale behind selecting the regional climatic gradient across the GVWSA, its three watersheds, and the key BEC zones (CWHxm, CWHmm, and CDF). The specific site-level details, such as topography, aspect, and detailed species lists for the shrub and herb layers at

Frog, Begbie, Swanson, and Worley Lakes, are important but relate more "methodological" in nature.

**[Material and Methods]**

The final paragraph (P8. L188~) is somewhat dense, and readers might need a more detailed explanation, particularly on how to reconstruct MAP. Additionally, if the paragraph becomes too long, the authors should clearly differentiate between what you reconstructed yourself (MAP from DWHI) and the data you extracted from published models (PALEO-PGEM and HadCM3) and other paleo-records (chironomid MJAT).

P6. L142. "A minimum of 300 grains per slide were" to "A minimum of 300 grains per slide was"

**[Results]**

Although the results data are comprehensive, the text often reads like a list of observations. To better engage readers, the authors should directly incorporate interpretations and key messages into the results section. Each subsection on pollen and charcoal at the sites would benefit from beginning with a clear, single-sentence summary highlighting the main findings. Furthermore, the narrative can be improved by reducing the taxonomic details in the main text. Overwhelming lists of minor taxa might be summarized in a phrase, such as "along with minor contributions from broadleaf taxa such as Acer and Quercus," with full details moved to supplementary diagrams. This allows the main text to focus on the main drivers of change. Finally, contrasts between functional groups should be highlighted as a story, not just listed. For example, instead of simply stating percentages, frame them: "This vegetation shift is reflected in the fire-adapted functional groups, which show a clear decline in 'invaders' (71% to 42%) and a rise in 'avoiders' (5% to 23%), consistent with a closing forest canopy."

The synthesis in Section 3.6 is the cornerstone of your Results and must be very clear to effectively present the overall story of the paper. To improve this, the authors should start by stating the main finding up front. After that, you need to go beyond merely reporting statistical values and include interpretation in simple language. For example, instead of just listing H and p-values, explain that the significant Kruskal-Wallis test indicates FRIs differ across sites and time, with follow-up pairwise tests showing the large increase at Worley Lake as the leading cause. Furthermore, the description of the composite charcoal curve should be clarified to highlight its key point: an early Holocene period of increased regional biomass burning followed by a general decline. Additionally, the climate data, although well-presented, can be more effectively integrated by explicitly connecting it to your fire records. Clearly state that the climatic shift to a cooler, wetter late Holocene aligns with your pollen-based precipitation reconstructions and is supported by independent model outputs and temperature proxies.

P.19 L343. "a pattern of asynchroneity" to "a pattern of asynchrony"

**[Discussion]**

Although the data in this section is comprehensive, the main arguments and their implications could be presented more strongly and clearly. The manuscript would benefit from explicitly stating the central finding at the very beginning of the Discussion section, immediately providing context for the reader. Additionally, in Sections 4.1 and 4.2, the key conclusions for each period should be clearly summarized at the start of each section. This would make the

main points more obvious and easier to understand, rather than requiring the reader to piece them together from the detailed evidence that follows.

The discussion provides a robust analysis, but its impact could be improved by combining ideas to enhance clarity and rhetorical strength. Currently, the site-by-site vegetation descriptions, while detailed, may overwhelm the reader. An integrated approach that clearly highlights the strong east-west gradient in vegetation openness would be more effective. This concept of synthesis also applies to the fire regime conclusions. The key finding that an east-west gradient was not evident in the early Holocene but became more pronounced later should be emphasized more prominently. Lastly, the various drivers proposed for late-Holocene fire variability represent a key complexity. Their influence would be stronger if organized into a single, focused paragraph that explores how these factors interacted as the dominant influence of a uniformly warm and dry climate started to weaken.

The analysis in section 4.3 can be enhanced by highlighting key insights. First, for the dry eastern sites, the main finding suggests that past and future changes are more influenced by shifts in fire severity and fuel connectivity than fire frequency. The paleodata show these sites have always experienced fire, hinting at a possible future with more frequent, lower-severity fires, but with a notable risk of high-severity events during climatic transitions, as evidenced by the period 500-1500 cal BP. Second, the moist western sites are most at risk of a complete regime shift. The record from Worley Lake acts as a key analogy, showing it was once as fire-prone as the eastern sites are today; this emphasizes that no part of this landscape is immune to major change, which is important for management strategies. Lastly, the influence of humans, from Indigenous burning to current suppression efforts, must be considered. Briefly explaining how human impacts interact with the climate-driven baseline will lead to a more complete and practical conclusion.

**[Conclusion]**

While the distinction between top-down and bottom-up controls is an important and insightful part of the authors' conclusion, its current presentation could be more impactful. The argument would be clearer if you replaced the hesitant phrasing and vague reference to "intrinsic factors" with a more direct and explicit explanation. Specifically, the main finding is the feedback loop the authors observed: climate acts as the primary top-down driver of initial vegetation change, which then alters fuel structures in a bottom-up manner. This bottom-up change in fuels significantly modifies the fire regime itself, demonstrating a tightly coupled and dynamic system.

The management implications are relevant but would be more compelling if linked more directly and authoritatively to your specific findings. The discussion of fire suppression, while important, currently feels somewhat disconnected from the paleodata just presented. To improve this, the authors could clearly frame the management implications through the lens of the two fire regime types they identified. This study offers essential long-term context for understanding what defines a climate-driven fire regime across various landscapes; the conclusion should emphasize this unique contribution to move beyond general statements and provide concrete, evidence-based guidance.

---

## Referee Comment (RC2)

[referee-annotated manuscript omitted]